# The impact of village heads' educational levels on adolescent academic performance: Evidence from rural China

Jing Li[1☯¤a]*, Huan Deng[2☯¤a], Jun Li[3¤b]

1 School of Management, Hunan Institute of Engineering, Xiangtan, Hunan, China, 2 School of Clinical Medicine, Xiangtan Medicine and Health Vocational College, Xiangtan, Hunan, China, 3 School of Business, Hunan Normal University, Changsha, Hunan, China

☯ These authors contributed equally to this work.
¤a Current address: Panlong Yuheyuan District, Yutang District, Xiangtan City, Hunan Province, China
¤b Current address: Hunan Normal University Erliban Campus, Yuelu District, Changsha City, Hunan Province, China
* lijingcssci@foxmail.com

**Data Availability Statement:** Our data is stored in figshare: https://doi.org/10.6084/m9.figshare.27194217.v3. Freely available.

**Funding:** This study was funded by the Hunan Province Education Science "14th Five-Year Plan"

## Abstract

This study investigates the relationship between the educational level of village heads and the academic performance of adolescents, using data from the China Family Panel Studies (CFPS). The analysis reveals that village chiefs with well-educated significantly enhance the academic outcomes of adolescents within their communities. This positive effect remains robust even after controlling for endogeneity through instrumental variables and conducting various robustness checks. Further investigation shows that these well-educated village leaders contribute to an increased provision of public goods, thereby improving the village's external environment, which in turn supports academic performance. Additionally, well-educated village chiefs serve as role models within the community's social network, positively influencing parental educational aspirations and enhancing adolescents' academic results. Notably, the impact of well-educated village chiefs is more pronounced among girls and adolescents from low-income families, underscoring its significance in promoting gender equity in education and breaking cycles of intergenerational poverty.

## Introduction

Developing countries face significant educational disparities across various rural development indicators. Rural adolescents, in particular, often occupy the most disadvantaged positions [1]. The State of Global Learning Poverty: 2022 Update report reveals that 70% of 10-year-olds are in learning poverty, unable to read and understand a simple text. This situation can be attributed to significant disparities in access to educational resources [2, 3] and family educational support [4]. Additionally, the exodus of rural elites exacerbates these challenges [5]. The growth environment of children, especially in villages, is widely regarded as a crucial factor influencing academic performance [6].

project [ND248771] to JL. The funders had no role in study design, data collection and analysis, decision to publish, or preparation of the manuscript.

**Competing interests:** The authors have declared that no competing interests exist.

In light of these disparities, the role of village heads in rural areas of developing countries becomes increasingly significant. Village heads serve as the primary managers of village governance and play a vital leadership role [7]. They are responsible not only for daily management and resource allocation but also for influencing villagers' educational attitudes, lifestyles, and social behaviors through the exchange of information and knowledge with other community members [8, 9]. This transmission of information can enhance villagers' appreciation for education and create a better learning environment for adolescents [10, 11]. Research has shown that the education level of village heads is directly related to their governance capabilities. Those with a strong educational background tend to implement policies, organize resources, and facilitate the dissemination of knowledge more effectively [12, 13]. Nonetheless, research on how the educational levels of village heads influence rural adolescents' education through governance mechanisms remains limited.

Leveraging the unique context of rural education in China, this study aims to analyze the potential impact of village heads' educational levels on family behavior and adolescents' academic performance. Addressing this question is crucial for several reasons. First, village heads play a key role in rural governance. Governments often rely on them to promote infrastructure development, social governance, and the provision of public services, all of which are essential steps in driving rural economic growth. Many rural areas in developing countries face governance challenges similar to those encountered in the early histories of developed nations [1, 2]. Existing literature highlights the significant role of village heads' knowledge and education in promoting rural economic development, increasing agricultural output, and raising farmers' incomes [6, 7, 9, 12–16]. However, there is ongoing debate about whether well-educated village heads are more favorable for enhancing the supply of public goods during the resource allocation process [14, 16–19].

In light of these considerations, this study investigates the impact of village heads' educational levels on adolescents' academic performance in rural China. It specifically examines how the educational attainment of village leaders influences the allocation of collective resources and shapes family educational expectations. This analysis seeks to elucidate the governance mechanisms at play, emphasizing the pivotal role of village heads in influencing educational outcomes. Understanding this relationship is crucial, as it addresses broader implications for rural development and social equity. Ultimately, this study aims to contribute to the enhancement of educational practices and policies in disadvantaged rural areas by underscoring the significance of effective leadership in promoting adolescent academic performance.

## Methods

### Research design

We used a cross-sectional study design to examine the association between village heads' educational levels and adolescents' academic performance, utilizing data from the CFPS to assess how these educational levels influence academic achievements.

### Study subjects and data sources

**Study subjects.** This study focuses on adolescents and their families in rural areas of China. In this context, "rural" specifically refers to rural communities in China, which often face challenges such as insufficient educational resources, relatively slow economic development, and a lack of social services. Adolescent education is regarded as a key factor in rural development, particularly in the broader context of reducing the urban-rural income gap, improving the well-being of rural residents, and promoting sustainable development.

Understanding the role of village heads and their influence on education is especially important. Village heads, as grassroots leaders in rural areas, are typically elected by local residents and are responsible for managing village affairs, promoting community participation in economic and social activities, and implementing national and local policies. The educational background of village heads is directly related to their governance capabilities, particularly regarding resource management, policy communication, and community collaboration. In this study, we specifically focus on the educational levels of village heads.

**Data sources.** Data were obtained through the official channels of the CFPS, which employs a stratified random sampling method to ensure the representativeness of the sample. The survey questionnaire, used as part of the CFPS, includes multiple-choice questions and scales that cover information on family background, educational environment, and characteristics of village heads. The CFPS is a nationwide, large-scale, multidisciplinary social tracking survey project that spans 25 provinces, municipalities, and autonomous regions in China. There is a large body of literature on scientific research using CFPS data [20, 21]. As of now, CFPS has released data from the years 2010, 2012, 2014, 2016, 2018, and 2020; however, only the 2010 and 2014 datasets include community/village-level data. Therefore, in accordance with the research objectives, this paper selects the 2014 CFPS data for empirical analysis. The 2014 CFPS dataset includes basic characteristics, such as the economic and social conditions of 365 villages, as well as demographic information about the village heads, along with data on the academic performance of adolescents within these villages. After retaining rural and adolescent samples and removing missing values for various variables, a total of 1,720 valid samples were included in the analysis.

## Variable definitions

**Educational level of village heads.** The educational level of village heads is the main explanatory variable in this paper, measured using the Educational Level of Village/Community Heads from the CFPS community/village questionnaire. This variable is ordinal, with values ranging from 1 to 6, corresponding to illiterate/semi-literate, primary school, middle school, high school, junior college, and undergraduate or higher education levels.

**Academic performance of adolescents.** This paper uses the average scores of the word group test and mathematics test from the child cognitive module in CFPS data to measure the academic performance of adolescents. The advantages of this approach include the uniformity of the word group test and mathematics test in CFPS data, which ensures the same level of difficulty, and the objective nature of the test questions, which ensures consistent scoring standards. However, children with higher levels of education may achieve higher test scores. To address this, the paper standardizes test scores by converting them into z-scores based on each adolescent's current educational level, aiming to reduce differences between groups. It is worth noting that to ensure the scientific validity and comparability of test scores, the CFPS team only tests individuals aged 10 years and older; the CFPS children's questionnaire only includes children under the age of 16. Therefore, the sample interval for adolescents in this paper is 10–15 years old. It is worth noting that standardization is performed under the original sample of rural families without excluding missing variables. Therefore, the standard deviation of the Academic performance variable in empirical analysis is not 0.

**Mechanism variables.** This paper explores how the educational level of village heads affects the academic performance of adolescents within the village from the perspective of the external village environment. Specifically, we examine two channels: one is the provision of public goods in the village, and the other is the role of the village head as a role model.

This paper uses the expenditure on public services (exp_public) within the village to measure the supply of public goods, which is treated as a continuous variable and log-transformed. Additionally, a poor and disorganized living environment can impact adolescents' cognitive abilities to some extent and can negatively model behavior, thus affecting academic performance. Therefore, we further discusses whether well-educated village heads can enhance adolescent academic performance through the management of environmental pollution in village living environments. The cleanliness of village roads is used as a proxy variable for the management of the village living environment. This is a categorical variable with values ranging from 1 to 7, where higher values indicate cleaner roads.

The role model effect's inherent manifestation in education is the motivation and expectation for education. This paper uses parents' educational expectations for adolescents to measure the role model effect. Educational expectations are measured using the CFPS question "Desired Level of Education for Children," with values ranging from 1 to 8, corresponding to no schooling necessary, primary school, middle school, high school, junior college, bachelor's degree, master's degree, and doctorate.

**Covariates.** In order to control for confounders, the following variables were included in the analysis, which were chosen a *priori* on a theoretical basis. First, the gender of the village head is included (male = 1; female = 0), as it may influence preferences for economic investment and public services [22]. Additionally, female village heads may exhibit more intergenerational care during village governance [11]. According to human capital theory [23], the age of the adolescent is a crucial factor affecting performance on the CFPS academic tests; therefore, it is included as a covariat. Furthermore, adolescents are influenced by external environments that vary by age and gender, which is why the gender of the adolescent is also controlled for. At the parental level, parental education is directly linked to adolescent academic performance, while parents' political status can provide access to better educational resources through political capital [24]. Hence, this paper controls for the highest level of education attained by parents (with values ranging from 1 to 6, similar to the educational level of the village head) and the highest political status (party member = 1; otherwise = 0).

Regarding family characteristics, family income significantly impacts adolescents' access to educational resources. In rural areas, insurance and land assets influence family education decisions [25, 26]. Additionally, the income risk associated with agricultural production can restrict educational investments, while non-farm employment involving labor migration may adversely affect adolescents [27]. Previous studies indicate that internet access can also impact the quantity and quality of education, as well as the accumulation of new human capital [28]. Moreover, increases in the number of health shocks and working family members can affect parental involvement and family education investment [23], thereby impacting adolescent academic performance. Consequently, this paper controls for various family characteristics, including family income (logarithmic), land assets (logarithmic), nature of family production (agricultural production = 1; otherwise = 0), internet access at home (yes = 1; no = 0), the number of family health shocks, and the number of working family members. Descriptive statistics of the variables are shown in Table 1.

## Data analysis methods

This study employs multiple regression analysis to examine the impact of village heads' educational levels on adolescents' academic performance, utilizing mechanism analysis to investigate the underlying pathways of this influence. To address potential endogeneity, this study will utilize instrumental variable methods and perform a series of robustness checks to ensure the reliability and validity of the findings.

**Table 1. Descriptive statistics of variables.**

| Variable | Obs | Mean or Pct | S.D | Min | Max |
|---|---|---|---|---|---|
| Academic performance | 1 720 | -0.12 | 0.90 | -2.54 | 2.21 |
| Educational Level of Village Heads | 1 720 | 3.56 | 0.91 | 1 | 6 |
| Gender of Village Director | 1 720 | 98% | - | 0 | 1 |
| Teenage age | 1 720 | 12.42 | 1.737 | 10 | 15 |
| Teenage Gender | 1 720 | 52% | - | 0 | 1 |
| The highest educational level of parents | 1 720 | 2.16 | 1.03 | 1 | 6 |
| Political identity of parents | 1 720 | 10% | - | 0 | 1 |
| Household income (log) | 1 720 | 10.18 | 1.22 | 0.69 | 14.12 |
| Land assets (log) | 1 720 | 8.29 | 4.05 | 0 | 14.27 |
| Agricultural production | 1 720 | 82% | - | 0 | 1 |
| Internet | 1 720 | 36% | - | 0 | 1 |
| No. Health shock | 1 720 | 0.51 | 0.75 | 0 | 5 |
| No. Household workers | 1 720 | 2.18 | 1.09 | 0 | 8 |
| Exp_public | 1 269 | 1.39 | 1.63 | 0 | 9.2 |
| Pollution control | 1 720 | 4.41 | 1.49 | 1 | 7 |
| Parents' educational expectations | 1 720 | 5.46 | 1.27 | 0 | 8 |

**Baseline estimation model.** According to the research purpose of this article, the following benchmark econometric model is set:

$$Edu\_Score_i = \alpha_1 + \beta_2 Edu\_year_i + \beta_3 Control_i + \varepsilon_i \tag{1}$$

Among them, *Edu_Score* represents the average of the youth phrase test and math test, *Edu_year* represents the education level of the village head, *Control* represents various control variables at the individual characteristics, youth characteristics, and family level of the village head, and $\varepsilon$ represents the random error term. In this study, the regression analysis model used is based on several key assumptions. First, we assume that the relationship between the educational level of village heads and adolescent academic performance is linear, meaning that each unit change in educational level leads to a corresponding change in academic performance. Additionally, we assume that the error terms in the model are independent and homoscedastic, which implies that there is no autocorrelation or heteroscedasticity, ensuring the validity and robustness of the regression coefficients. At the same time, the CFPS data ensures the randomness and representativeness of the sample selection.

**Two-stage least squares model.** The sources of endogeneity primarily include reverse causality and omitted variables. Although the academic performance of adolescents does not affect the educational level of village heads, thereby eliminating the possibility of reverse causality, omitted variables remain a significant factor causing endogeneity in the baseline model. Therefore, this paper attempts to mitigate endogeneity issues using the instrumental variable (IV) approach. Wooldridge (2003) noted that group mean variables are commonly used to overcome the endogeneity of individual variables [29]. The paper employs the average educational level of village heads at the county level as the instrumental variable. On one hand, policy-wise, within the same area, the arrangement of village heads across different villages has a certain policy-relatedness [30]. The average education level of county-level village leaders reflects the characteristics of the education level of village leaders in the region as a whole, which is strongly correlated with the education level of individual village leaders. It has been shown in the literature that there is a correlation between regional characteristics such as social capital and regional governance level and individual leader characteristics. For example,

Moffitt (1993) notes that regional characteristic means are often effective in capturing changes in individual-level variables [31]. Case and Deaton (1999) demonstrated the relevance of the mean variable by using district educational resource means to study individual educational performance [32]. On the other hand, the average educational level of village heads in a county does not directly impact the academic performance of adolescents within a single village, satisfying the exclusivity of the instrumental variable. However, due to the CFPS data possibly having fewer samples for some counties' villages, this could lead to a weak instrumental variable problem. Therefore, following existing literature, the paper also relaxes the level of the instrumental variable to the provincial level, using the average educational level of village heads at the provincial level [30].

**Robustness testing methods.** To ensure our results remain consistent across different model specifications or sample selections, thereby verifying result stability and enhancing the persuasiveness of the study, this paper conducts robustness tests from the following perspectives. First, we redefine the educational level of village leaders using three binary dummy variables: college degree, high school diploma, and secondary school or below. This allows for a more nuanced representation of the educational backgrounds of village leaders. Second, we assess adolescent academic performance by estimating math and language scores separately, rather than relying solely on grade point average. Given the term limits and electoral fluctuations for village heads, changes in office may lead to variations in their educational levels. If such changes occur within a short time frame, they could introduce bias into the estimated results. To address this potential issue, the paper utilizes CFPS data from 2010 and re-estimates Eq (1) by retaining only those samples where the educational levels of village heads remained unchanged between 2010 and 2014. Finally, recognizing that adolescents' academic performances are correlated at the village level, the paper further clusters standard errors at the village level to account for any intra-village correlation

**Mechanism test model.** This section will further validate Hypothesis 2 using the three-step method of mechanism testing known as the BK (Baron and Kenny) method [33].

Step One involves examining the impact of the educational level of village heads on adolescent academic performance, as specified in Eq (2).

Step Two assesses whether there is a significant impact of the educational level of village heads on the mechanism variables, as specified in Eq (3). *Mech* Represents mechanism variables.

Step Three incorporates both the educational level of village heads and the mechanism variables into the model, as shown in Eq (4).

If, in Eq (4), the mechanism variable (*Mech*) is significant and the coefficient of the village head's educational level decreases, then the mechanism is considered valid. The equations are laid out as follows:

$$Edu\_Score_i = \alpha_1 + \beta_2 Edu\_year_i + \beta_3 Control_i + \varepsilon_i \tag{2}$$

$$Mech_i = \alpha_1 + \beta_2 Edu\_year_i + \beta_3 Control_i + \varepsilon_i \tag{3}$$

$$Edu\_Score_i = \alpha_1 + \beta_2 Edu\_year_i + \beta_3 Mech_i + \beta_4 Control_i + \varepsilon_i \tag{4}$$

This structured approach allows for a clear assessment of whether the proposed mechanisms (such as increased access to educational resources, improved public goods, or role model effects) mediate the relationship between the educational level of village heads and adolescent academic performance. By examining the significance and the size of the coefficients in these

equations, one can determine the extent to which these mechanisms play a role in influencing adolescent outcomes.

## Ethics statement

To ensure the maximum protection of the rights of participants in the project, the CFPS regularly submits ethical review or continuous review applications to the Biomedical Ethics Committee of Peking University. Data collection is conducted only after receiving ethical approval. The ethical review approval number for the CFPS project is uniformly IRB00001052-14010 and remains consistent across different survey waves. We obtained informed written consent from all participants.

# Results

## Benchmark regression results

Table 2 presents the Ordinary Least Squares (OLS) regression results regarding the influence of village heads' educational levels on adolescents' academic performance. Model 1 does not control for any variables, providing a straightforward analysis of the relationship between the educational level of village heads and adolescent academic performance. The results indicate a significant positive correlation at the 1% level, with a coefficient of 0.120.

To mitigate the impact of omitted variables and enhance the robustness of the model, Models 2 and 3 progressively incorporate individual characteristics of adolescents, parental characteristics, and family attributes. Results from Model 2 show that even after accounting for the individual traits of adolescents and parents—both of which significantly influence academic performance—the educational level of village heads remains positively correlated with adolescents' academic outcomes, albeit with a reduced effect size. In Model 3, after further controlling for family characteristics, the coefficient decreases to 0.096. Nevertheless, the educational level of village heads continues to exhibit a significant positive correlation with adolescent academic performance at the 1% level. Specifically, for each unit increase in the educational level of the village head, the standard deviation of adolescents' academic performance scores increases by 0.096.

## Addressing endogeneity issues

Table 3 presents the results of the Two-Stage Least Squares (2SLS) regression analysis. Models 4 and 5 showcase the first-stage regression results, employing the average educational levels of village heads at the county and provincial levels as instrumental variables, respectively. The findings indicate that both instrumental variables are significantly positively correlated with the educational levels of village heads, thus confirming their relevance. Furthermore, the weak instrumental variable tests reported at the bottom of Models 4 and 5 demonstrate C-D Wald F-statistics significantly exceeding 16.38, effectively ruling out the possibility of weak instrumental variables.

Models 6 and 7 provide the second-stage regression results of the 2SLS analysis. It is clear that whether using the average educational level of village heads at the county or provincial level as the instrumental variable, the educational level of village heads remains significantly positively correlated with adolescents' academic performance at the 1% level.

## Robustness tests

**Redefining the educational level of village heads.**   The regression results presented in Table 4, Models 8 to 10, demonstrate that village heads with a university degree significantly positively influence adolescents' academic performance. The coefficient for university

**Table 2. Education level of village directors and academic performance of adolescents.**

| Variable | Model 1 | Model 2 | Model 3 |
|---|---|---|---|
| | Academic performbmance | | |
| Edu_year | 0.120** | 0.106** | 0.096** |
| | [0.075,0.165] | [0.065,0.147] | [0.055,0.137] |
| Gender of Village Director | | 0.019 | 0.018 |
| | | [−0.030,0.068] | [−0.029,0.065] |
| Teenage age | | 0.228** | 0.228** |
| | | [0.205,0.251] | [0.206,0.250] |
| Teenage Gender | | -0.093* | -0.086* |
| | | [−0.167,−0.019] | [−0.160,−0.012] |
| Highest_edu of parents | | 0.133** | 0.106** |
| | | [0.096,0.170] | [0.069,0.143] |
| Party_parents | | -0.106 | -0.096 |
| | | [−0.230,0.018] | [−0.221,0.029] |
| Household income | | | 0.054** |
| | | | [0.023,0.085] |
| Land assets | | | -0.011 |
| | | | [−0.025,0.003] |
| Agricultural production | | | 0.036 |
| | | | [−0.122,0.194] |
| Internet | | | 0.167** |
| | | | [0.083,0.251] |
| No. Health shock | | | -0.022 |
| | | | [−0.079,0.035] |
| No. Household workers | | | -0.044* |
| | | | [−0.087,−0.001] |
| Observations | 1 720 | 1 720 | 1 720 |

[a]** stands for p<0.01, estimated coefficient significant at 1% level

* stands for p<0.05, estimated coefficient significant at 5% level.

[b]Numerical intervals in square brackets are 95% confidence intervals.

**Table 3. Endogeneity problem handling.**

| Variable | Model 4 | Model 5 | Model 6 | Model 7 |
|---|---|---|---|---|
| | First stage | | Second stage | |
| | Edu_year | | Academic performance | |
| Edu_year | | | 0.177** | 0.223** |
| | | | [0.115,0.226] | [0.110,0.359] |
| IV1 | 0.974** | | | |
| | [0.927,1.067] | | | |
| IV2 | | 0.879** | | |
| | | [0.714,1.021] | | |
| C-D Wald Fstatistic | 1688.73 | 202.169 | | |
| Observations | 1 720 | 1 720 | 1 720 | 1 720 |

[a]** stands for p<0.01, estimated coefficient significant at 1% level

* stands for p<0.05, estimated coefficient significant at 5% level.

[b]Numerical intervals in square brackets are 95% confidence intervals.

**Table 4. Robustness test.**

| Variable | Model 8 | Model 9 | Model 10 | Model 11 | Model 12 | Model 13 | Model 14 |
|---|---|---|---|---|---|---|---|
| | Academic performance | | | mathematics | language | Academic performance | |
| Edu_year | | | | 0.0833** | 0.109** | 0.159** | 0.096** |
| | | | | [0.0384,0.128] | [0.0601,0.158] | [0.0767, 0.2413] | [0.0254, 0.1666] |
| University degree | 0.131** | | | | | | |
| | [0.0356,0.226] | | | | | | |
| High school education | | 0.073 | | | | | |
| | | [-0.003,0.149] | | | | | |
| Junior high school and below | | | -0.129** | | | | |
| | | | [-0.204,-0.055] | | | | |
| Control variables | YES | YES | YES | YES | YES | YES | YES |
| Observations | 1 720 | 1 720 | 1 720 | 2 868 | 2 871 | 624 | 1 720 |

[a]** stands for p<0.01, estimated coefficient significant at 1% level

* stands for p<0.05, estimated coefficient significant at 5% level.

[b]Numerical intervals in square brackets are 95% confidence intervals.

education is 0.131, which is notably higher than the coefficient for high school education at 0.073. This indicates that higher educational attainment among village heads corresponds to a greater positive impact on adolescents' academic performance.

**Replacing the variable for adolescent academic performance.** The results are presented in Table 4, Models 11 and 12, respectively. The estimated coefficients for both variables are significantly positive at the 1% level.

**Considering changes in village heads.** The results in Model 13 indicate that even after accounting for changes in village heads, their educational level continues to significantly impact adolescents' academic performance at the 1% level.

**Adjusting clustered standard errors.** The results after adjusting for clustered standard errors, as shown in Model 14, remain robust.

## Mechanism analysis

**Public goods supply.** Table 5 presents the regression results for this section. Models 15 and 16 re-estimate the impact of the village head's educational level on adolescent academic performance, taking into account missing values in the mechanism variables. Models 17 and 18 examine the effect of the village head's educational level on the supply of public goods and environmental pollution management, with both showing p-values less than 0.01. The results in Models 19 and 20 indicate that the supply of public goods and environmental pollution management in the village significantly influence adolescent academic performance; however, the coefficient for the village head's educational level is lower compared to those in Models 15 and 16.

**Role model effect.** Models 21 to 23 of Table 6 report the results of the tests for the role model effect mechanism. These models correspond to the first, second, and third steps of the three-step method, respectively. The educational expectations are significantly positively correlated with adolescent academic performance at the 1% level in Model 23, with a noticeable decrease in the coefficient of the village head's educational level compared to Model 21.

## Heterogeneity analysis—adolescent gender and family income

Models 24 and 25 in Table 7 report the regression results segmented by adolescent gender. The findings indicate that girls are more significantly influenced by the educational level of

**Table 5. Mechanism analysis of village residential environment governance.**

| Variable | Model 15 | Model 16 | Model 17 | Model 18 | Model 19 | Model 20 |
|---|---|---|---|---|---|---|
| | Academic performance | | Public goods supply | Pollution control | Academic performance | |
| Edu_year | 0.111** | 0.096** | 0.223** | 0.146** | 0.105** | 0.092** |
| | [0.0659, 0.1561] | [0.0558,0.1362] | [0.123,0.323] | [0.066,0.226] | [0.058,0.152] | [0.051,0.133] |
| Pg supply | | | | | 0.029* | |
| | | | | | [0.005,0.053] | |
| C-Pollution | | | | | | 0.030* |
| | | | | | | [0.002,0.058] |
| C-vars | YES | YES | YES | YES | YES | YES |
| Observations | 1 291 | 1 720 | 1 291 | 1 720 | 1 291 | 1 720 |

[a]** stands for p<0.01, estimated coefficient significant at 1% level

* stands for p<0.05, estimated coefficient significant at 5% level.

[b]Numerical intervals in square brackets are 95% confidence intervals.

**Table 6. Mechanism analysis of example effect.**

| Variable | Model 21 | Model 22 | Model 23 |
|---|---|---|---|
| | Academic performance | Educational expectations | Academic performance |
| Edu_year | 0.098** | 0.101** | 0.085** |
| | [0.057,0.139] | [0.040,0.162] | [0.046,0.124] |
| Educational expectations | | | 0.129** |
| | | | [0.096,0.162] |
| Control variables | YES | YES | YES |
| Observations | 1 712 | 1 712 | 1 712 |

[a]** stands for p<0.01, estimated coefficient significant at 1% level

* stands for p<0.05, estimated coefficient significant at 5% level.

[b]Numerical intervals in square brackets are 95% confidence intervals.

**Table 7. Heterogeneity analysis.**

| Variable | Model 24 | Model 25 | Model 26 | Model 27 | Model 28 |
|---|---|---|---|---|---|
| | Academic performance | | | | |
| | Male | Female | Low | Middle | High |
| Edu_year | 0.083** | 0.096** | 0.160** | 0.040 | 0.084* |
| | [0.024,0.142] | [0.039,0.153] | [0.078,0.242] | [−0.013,0.093] | [0.004,0.164] |
| Control variables | YES | YES | YES | YES | YES |
| Observations | 900 | 820 | 430 | 859 | 431 |

[a]** stands for p<0.01, estimated coefficient significant at 1% level

* stands for p<0.05, estimated coefficient significant at 5% level.

[b]Numerical intervals in square brackets are 95% confidence intervals.

village heads compared to boys. The mechanism analysis reveals that well-educated village heads can enhance the village living environment by increasing the supply of public goods.

Models 26 to 28 present the model estimation results after dividing the sample into low-income, middle-income, and high-income families based on the 25th and 75th percentiles of income. For low-income families, the educational level of village heads significantly influences adolescent academic performance at the 1% level, with a coefficient of 0.160. In high-income families, highly educated village heads also positively impact adolescent academic performance, with a coefficient of 0.084; however, this is lower than the effect observed in low-income families. Notably, in middle-income families, there is no significant correlation.

## Discussion

In this study, we examine how the educational levels of village heads affect adolescents' academic performance in China, a major agricultural nation, and highlight the village head's role in rural educational governance. OLS regression analysis reveals a significant correlation between village heads' education and adolescents' academic achievements. This aligns with previous research, indicating that educated leaders can improve educational outcomes. For instance, Lahoti and Saho (2020) [34] found that educated political leaders enhance their constituents' educational results. Karadag's (2020) meta-analysis of studies from 2008 to 2018 also shows a significant impact of educational leadership on student performance [35]. Our study uniquely highlights the external benefits of village leaders' educational levels on adolescents, emphasizing that village communities play a crucial role in influencing educational outcomes, in addition to schools and families. This underscores the importance of village heads as key factors in governance, with the potential to drive long-term benefits for adolescents' academic success and the overall quality of rural education. These findings provide insights for policy discussions, emphasizing the need for greater investment in rural governance resources. Additionally, they contribute to understanding the impacts of public education policy expansion in developing countries and impoverished areas.

Our results indicate that the educational level of village heads not only affects academic performance but also influences adolescents through improved community environments and elevated educational expectations within families. This highlights the significant role of the external environment in villages. Our empirical analysis aligns with existing literature; for example, Jain et al. (2023) [5] found that educated leaders enhance the provision of roads, electricity, and power. Additionally, Zhang et al. (2024) [13] discovered that highly educated village heads can promote local infrastructure development. Existing research also indicates that the neighborhood environment in urban areas significantly impacts children's education [36–40]. Specifically, improvements in community resources, environmental quality, social networks, and public safety during childhood enhance academic performance [41–47]. Unlike studies that focus on urban communities, our research examines rural communities, emphasizing the critical role they play in educational outcomes. In developing countries, rural education faces severe challenges, particularly with high rates of left-behind children, which directly impacts poverty alleviation and social equity. Addressing these issues is vital for fostering equitable educational opportunities and improving overall community well-being.

The strong positive correlation between parents' educational expectations and adolescents' academic performance, particularly evident in Model 23, underscores the transformative power of community leadership as a role model. This observation aligns with literature highlighting the influence of community leaders on parental attitudes toward education. For instance, Beaman et al. (2012) [10] demonstrated that female leadership positively impacts girls' aspirations and educational attainment by comparing villages in India with randomly

assigned female leaders. They argue that female leaders primarily effect this change by serving as role models, thus elevating both girls' self-expectations and parental expectations for their daughters. Our findings complement existing research by showing that it is not only the educational levels of school leaders that significantly impact adolescents' educational outcomes [48, 49], but that community leadership also plays a crucial role. These findings provide actionable insights for policymakers, indicating that investing in the education of local leaders can significantly enhance community education standards—not just through direct policy enforcement, but also by serving as effective role models and setting positive norms.

Finally, our empirical results indicate that the educational levels of village heads have heterogeneous effects on adolescents' academic performance. The results in Table 7 reveal gender-based differences in this influence, showing a more pronounced impact on girls than on boys. This differential effect may stem from the village heads' role in improving the living environment through enhanced public goods provision, which tends to benefit female adolescents more significantly. This finding aligns with research indicating that girls are more sensitive to improvements in their immediate environments [50]. The heterogeneity analysis across different income groups provides nuanced insights. For low-income families, the educational level of village heads positively impacts adolescents' academic performance, highlighting the critical role that village governance can play in compensating for a lack of educational resources in economically disadvantaged settings. Similar to the findings of Zhang et al. (2024) [13], highly educated village heads can alleviate poverty in the community. Interestingly, the analysis shows no significant correlation for middle-income families, possibly indicating a threshold effect where improvements in governance and public goods provision do not substantially alter educational trajectories, given that these families have adequate resources and a lesser reliance on public goods. These findings underscore the importance of targeted policy interventions. Educational policies and community development initiatives should be tailored to address the specific needs and existing resources of different demographic groups.

The results of this study are based on stratified sampling data from rural villages in China. While we draw several conclusions, it is important to acknowledge the limitations due to data constraints, which prevent a comprehensive capture of all possible influencing factors. Notably, in rural China, village heads are elected by local residents, and their authority in the community may outweigh their educational leadership. Future research should carefully consider the impacts of cultural and social structures in this context.

## Conclusion

Our study reveals that the educational level of village heads significantly influences rural economic growth and social governance, with intergenerational effects on human capital accumulation. Village heads with higher educational attainment substantially enhance adolescents' academic performance in their communities through improved public goods provision and by serving as role models that elevate parental educational expectations. These findings suggest that investing in the education of village leaders is a crucial strategy for promoting sustainable rural education and achieving broader revitalization goals. Additionally, future research should focus on the impact of village heads with higher educational qualifications, such as those from the 'College Student Village Official' program (a national initiative that appoints university graduates as village leaders), as well as the effects of different academic backgrounds on adolescents' academic performance. Furthermore, emphasizing the long-term influence of these leaders could provide deeper insights into the benefits of effective educational leadership in rural communities.

## Author Contributions

**Conceptualization:** Huan Deng.

**Data curation:** Jing Li, Huan Deng.

**Formal analysis:** Jing Li.

**Funding acquisition:** Jing Li.

**Investigation:** Jing Li, Huan Deng.

**Methodology:** Huan Deng, Jun Li.

**Project administration:** Huan Deng, Jun Li.

**Resources:** Huan Deng, Jun Li.

**Software:** Jing Li, Jun Li.

**Supervision:** Jing Li, Jun Li.

**Validation:** Jing Li, Jun Li.

**Visualization:** Jun Li.

**Writing – original draft:** Jing Li, Huan Deng, Jun Li.

**Writing – review & editing:** Jing Li, Jun Li.

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
