## [Decision Letter · Decision Letter 0]

30 Aug 2024

PONE-D-24-18436Educational Attainment of Village Heads and Adolescents' Academic Performance - A Perspective Based on the External Environment of Village SettlementsPLOS ONE

Dear Dr. Li,

Thank you for submitting your manuscript to PLOS ONE. After careful consideration, we feel that it has merit but does not fully meet PLOS ONE’s publication criteria as it currently stands. Therefore, we invite you to submit a revised version of the manuscript that addresses the points raised during the review process.

We look forward to receiving your revised manuscript.

Kind regards,

Fabiola Vincent Moshi

Academic Editor

PLOS ONE

Journal Requirements: When submitting your revision, we need you to address these additional requirements. 1. Please ensure that your manuscript meets PLOS ONE's style requirements, including those for file naming. The PLOS ONE style templates can be found at https://journals.plos.org/plosone/s/file?id=wjVg/PLOSOne_formatting_sample_main_body.pdf and https://journals.plos.org/plosone/s/file?id=ba62/PLOSOne_formatting_sample_title_authors_affiliations.pdf 2. Please include your full ethics statement in the ‘Methods’ section of your manuscript file. In your statement, please include the full name of the IRB or ethics committee who approved or waived your study, as well as whether or not you obtained informed written or verbal consent. If consent was waived for your study, please include this information in your statement as well. 3. Please include a copy of Table 7 which you refer to in your text on page 33 in PDF submission. 

**Additional Editor Comments:**

Include strengths and limitations of the study

Reviewers' comments:

Reviewer's Responses to Questions

**Comments to the Author**

1. Is the manuscript technically sound, and do the data support the conclusions?

Reviewer #1: Partly

Reviewer #2: Yes

2. Has the statistical analysis been performed appropriately and rigorously? 

Reviewer #1: I Don't Know

Reviewer #2: Yes

3. Have the authors made all data underlying the findings in their manuscript fully available?

Reviewer #1: Yes

Reviewer #2: Yes

4. Is the manuscript presented in an intelligible fashion and written in standard English?

Reviewer #1: No

Reviewer #2: Yes

5. Review Comments to the Author

Reviewer #1: Generally, this manuscript though well intended, is seemingly deficient in several areas. To this end the author is best advised to consider having a section in the manuscript devoted to the study population and giving us a clear background of what it is about the population that is special and warrants it being investigated by way of this study.

What would help would be for the author to review the introduction section, section 1. This section could be improved if the author seeks to ensure that there is a better level of coherence in the thoughts being expressed.

The manuscript is not clear in stating the economic or social platform that is used for information transfer or knowledge sharing by village elders to young persons in the population.

Assertions and suppositions, if not opinions, ought to be supported by appropriate citations/references/literature. This sadly was missing on several instances throughout the manuscript. The author needs to give readers a better understanding of certain characteristics of the study population. What does rural mean in the context of the study? What does knowledge of the village elders speak to? Do they have university degrees etc.? what is it about them that is distinct and lends to the arguments being made?

The author needs to establish coherence. There are several instances where thoughts are disjointed and appear out of place. This is very much noticeable in the Sections 1 and 2 of the manuscript.

The author would be best advised to make use of the funnel approach when constructing section 1/or revising same.

Additionally, this section lacks reference to any statistical/data to support some of the assertions made. One is not sure of the origin/source of certain statements.

There are instances where areas in section 1, 2 and 3 are not well developed or require further expansion. There are too many disconnected parts.

In section 4 (e.g section 4.4.1) of the manuscript, the author is often guilty of beginning to speak of a method of measurement/analysis but then in the same breath talks of a discussion. This disjointedness ought to be addressed. An explanation of a methodology cannot suddenly descend to a discussion of something else that appears unrelated.

Additional Comments

The manuscript was void of page numbers and line numbering. This made the process of tagging areas of concern rather difficult for the reviewer.

The article title could be more specific/definitive and self-explanatory without having to refer to the body of the manuscript should the author choose to identify the study population therein.

The author is best advised to choose a single citation style instead of using two distinct citation styles at the same time. This flaw is very prominent in the introduction section of the manuscript. In other areas of the manuscript the author is guilty of alternating between citation styles. The author should seek to be consistent by using a single citation style.

Certain sections of the manuscript appear to be disjointed and out of place. Evidence of this exists in the second paragraph of the 'Introduction' wherein, there is a sentence which starts with the word "Besides...' Additionally, the author is best advised to cite references when making statements and/or inferences that are not original to the author. Evidence of this is seen in the sentence in paragraph 2 of the 'Introduction' wherein there is a sentence that begins with 'therefore...'

In the final paragraph of the 'Introduction' the author introduced "besides the aforementioned policy implications..." This note in the opinion of the reviewer is best suited for the discussions or recommendations sections of the manuscript. Seeking to discuss policy implications at this juncture of the manuscript appears out of place. Additionally, the final sentence of the las paragraph of the "introduction" appears best suited for the section on 'research design/methodology' This sentence also appears to be out of place. The author is best advised to correct this.

Section 2 (Literature Review) is also replete with the use of inconsistent citation styles. In paragraph 2 of Section 2, the author attempts to introduce some four aspects of educational development of adolescents. It is apparent that the author made use of the incorrect punctuation mark after stating each of the four aspects. This made reading this section a bit of a challenge given repetitious use of words heretofore mentioned in the assumed heading sentences.(section 2.1). Additionally, having each of the four aspects in the same paragraph suggest that they are invariably connected when this is not necessarily the case. The author is advised to rephrase this paragraph so as to remove any ambiguity/doubts surrounding the information being conveyed. Each point could be placed in its own paragraph.

In section 2.2, the author made a bold statement on "talented people governing the village..." This point important as it might be seemingly is void of any credible citation. Additionally, it begs the question as to where exactly this might be occurring. Would need revisiting by the author

In the final paragraph of Section 2, the author in attempting to make a case in respect of "The nature of duties and powers... seemingly did not present a clear harmonization of thoughts. This made this point appear disjointed and out of place. This too warrants rephrasing in alignment with relevancy of the section it falls under.

Section 2.3, though a god showcase of the theoretical premise upon which the paper is written, again the author has been guilty of not citing correctly/not citing at all.

Under section 3 (Research Design), it would augur well for good reading should the author include an area to address the study population specifically.

Under section 3.2.2, it is felt that the first two sentences though a bold statement appear not to be cited. In the absence of citations here, the question asked is whether the author is presenting an opinion or a mere conjecture.

Within the first paragraph of section 3.2.2, the author suddenly speaks of an econometric model and standardization, words of which connotes methods of analysis. Perhaps the author may opt to include a section that addresses specifically what these forms of analyses entail. Having a section specifically devoted to the methodology/methods employed could address this. This section could also include the kinds and types of analyses needed instead of morphing analysis with results.

Under section 3.2.3, again the author though referring to ''drawing on literature" fails to at the least cite one literature that that adopts the approach of controls for gender in the manner expressed in the manuscript. This is a n area for improvement by the author. the author is advised to at the least provide a basis/source for certain statements made or being alluded to.

Under section 4 the author appears to make reference to several kinds of analyses underpinning the results. However, what is not entirely clear is what are some of the key assumptions upon which these analyses are based. This need to be borne out in this section. Additionally, re: paragraph that speaks of stepwise regression, the author in closing attempts to morph results with discussion/implications of work. The author is best advised to include this in the discussion/conclusion section of the manuscript.

The contents in each section should be distinctly related to the titled of the given section heading.

Section 4.2 appear to be properly presented.

All other areas of the manuscript appear to be well constructed, with use of the appropriate forms of analyses and arriving at the appropriate results.

Section 5 generally appears to be sufficiently succinct with the author using the study findings to make appropriate recommendations.

Other areas for improvement

Review the need for three significant levels when presenting results in tabular form. What is the basis for this? Must be supported by proper citation/literature.

Reviewer #2: General comment: I commend the authors for their innovative research ideas. However, the manuscript is quite lengthy and would benefit from a more conventional structure (e.g., abstract, introduction, methods, results, discussion, and conclusion). Adopting this format and streamlining the content could enhance overall clarity and coherence, reducing potential confusion for readers.

Comment #1: I recommend consistently using the term 'China Family Panel Studies' throughout the abstract and methods sections rather than using 'China Family Tracking Survey’ in the abstract.

Comment#2: The manuscript contains a mixture of citation styles. I recommend using a single citation style consistently throughout the manuscript—either Vancouver or Harvard—according to the journal's guidelines.

Comment #3: The term 'highly educated' usually refers to individuals with advanced degrees, such as a PhD or master's, who have specialized knowledge and skills. Since village leaders in China are unlikely to hold such qualifications, I recommend using the term 'well-educated' throughout the manuscript instead. If the authors have a compelling justification for using 'highly educated,' they may retain it, but this should be clearly explained.

Comment#4: The literature review section is overly lengthy, and since this is not a systematic review, a concise summary would be more appropriate. I recommend merging the literature review with the introduction to create a focused and coherent introduction (specifically, 3 to 4 pages or less) that effectively summarizes the background, research context, and study hypothesis.

Comment#5: Table 1: Categorical variables, including gender, political identity, agricultural production, and internet use, should not be described using mean and standard deviation (SD), as these summary measures are appropriate only for continuous data with a normal distribution. Instead, categorical data should be presented using frequencies and percentages.

Comment#5: Tables: I recommend naming the columns as Model 1, Model 2, and Model 3 etc., instead of (1), (2), and (3).

Comment#6: Tables: I understand that the significance levels of 1%, 5%, and 10% correspond to p-values of 0.01, 0.05, and 0.1, respectively. However, most research considers associations with p-values below 0.05 as statistically significant. Therefore, treating p-values of 0.1 as indicating a significant association may be unconventional and should be carefully justified.

Comment#7: Tables: To better understand the data, it is more appropriate to present the coefficients and their 95% confidence intervals.

Comment#8: All tables should be self-explanatory. For example, in Table 5, Column 1: Edu_year 0.111***(0.023) – it's unclear what the numbers in brackets represent. Please clarify this in the table or the table notes.

Comment#9: I recommend restructuring the article to follow this sequence: abstract, introduction, methods, results, discussion, and conclusion. The current format presents the discussion before the results (i.e., in the literature review section), which can be confusing for readers. Additionally, combining the discussion with the results further adds to the confusion. A more precise separation of these sections would improve readability and coherence.

Comment#10: The conclusion is too lengthy. Including detailed discussions of mechanisms or extending the discussion detracts from the main points. This section should be concise, focusing exclusively on the study's major findings.

Comment#11: Policy recommendations: This section would be more effective if it informed the reader about the current policy and clearly explained how the study's findings could change, influence, or improve the existing policies. The current version lacks focus.

Comment#12: Adding line numbers to the manuscript would simplify the review process, allowing for more precise referencing and feedback.

6. PLOS authors have the option to publish the peer review history of their article (what does this mean?). If published, this will include your full peer review and any attached files.

Reviewer #1: No

Reviewer #2: No

---

## [Author Response · Author response to Decision Letter 0]

9 Oct 2024

Dear Editor and Reviewers,

Thanks very much for taking your time to review this manuscript. I really appreciate all your comments and suggestions! Please find my itemized responses in below and my revisions/corrections in the re-submitted files. 

We hope this revised manuscript has addressed your concerns, and look forward to hearing from you.

We have uploaded a "Revised Manuscript with Track Changes." However, we apologize for not using the track changes feature during the earlier revisions. Instead, we have utilized the comments function to explain the previously revised sections.

Aside from the revisions mentioned below, we have made the following changes: 1. The surname of the second author was corrected from 'Deng Huan' to 'Huan Deng'; 2. During the revision stage, we received funding support, and we have added this information in the funding section.

Please refer to the attachment “Response to reviewers” for instructions on how to make changes to the form.

Thanks again!

*Editor

1.Please ensure that your manuscript meets PLOS ONE's style requirements, including those for file naming. 

Response: Dear Editor, we have revised the manuscript according to the PLOS ONE style templates you provided.

2.Please include your full ethics statement in the ‘Methods’ section of your manuscript file. In your statement, please include the full name of the IRB or ethics committee who approved or waived your study, as well as whether or not you obtained informed written or verbal consent. If consent was waived for your study, please include this information in your statement as well.

Response: The complete ethics statement has been added to the "Methods" section, specifically in lines 149-155.

3.Please include a copy of Table 7 which you refer to in your text on page 33 in PDF submission

Response: We sincerely apologize for the oversight; Table 7 has been added on line 403.

*Reviewer 1

1.The manuscript is not clear in stating the economic or social platform that is used for information transfer or knowledge sharing by village elders to young persons in the population.

Response: We have added a sentence in the introduction (lines 51-59) to clarify the role of village heads, including their function in disseminating knowledge to the villagers.

2.Assertions and suppositions, if not opinions, ought to be supported by appropriate citations/references/literature. This sadly was missing on several instances throughout the manuscript. 

Response: Thank you for your valuable suggestion; we have supported our assertions and suppositions with relevant literature.

3.The author needs to give readers a better understanding of certain characteristics of the study population. What does rural mean in the context of the study? What does knowledge of the village elders speak to? Do they have university degrees etc.? what is it about them that is distinct and lends to the arguments being made?

Response: The introduction now elaborates on the role of village heads, particularly in lines 50-59. Additionally, we emphasize the importance of studying rural areas and the specific selection of our research subjects in the third paragraph. Your concerns are also addressed in the Methods section under "Study Subjects" (lines 118-130).

4.Additionally, this section lacks reference to any statistical/data to support some of the assertions made. One is not sure of the origin/source of certain statements.

Response: We have conducted a comprehensive revision of the introduction to ensure structural consistency. Following Reviewer 2's recommendation, we merged the introduction with the literature review, defining the significance of the research while providing a concise overview of key literature.

5.The author is best advised to cite references when making statements and/or inferences that are not original to the author. Evidence of this is seen in the sentence in paragraph 2 of the 'Introduction' wherein there is a sentence that begins with 'therefore...'

 Response: We have cited data from UNESCO in the introduction to underscore the severity of educational issues faced by students (line 43). Controversial assertions are now supported by appropriate references.

6.In the final paragraph of the 'Introduction' the author introduced "besides the aforementioned policy implications..." This note in the opinion of the reviewer is best suited for the discussions or recommendations sections of the manuscript. Seeking to discuss policy implications at this juncture of the manuscript appears out of place. Additionally, the final sentence of the las paragraph of the "introduction" appears best suited for the section on 'research design/methodology' This sentence also appears to be out of place. The author is best advised to correct this.

Response: For any contentious assertions, we have provided supporting literature. Thank you for your suggestion; we have rewritten this section accordingly. In line with Reviewer 2's advice, we merged the literature review with the relevant citations, actively addressing the corresponding issues during this process.

7.Section 2 (Literature Review) is also replete with the use of inconsistent citation styles. In paragraph 2 of Section 2, the author attempts to introduce some four aspects of educational development of adolescents. It is apparent that the author made use of the incorrect punctuation mark after stating each of the four aspects. This made reading this section a bit of a challenge given repetitious use of words heretofore mentioned in the assumed heading sentences.(section 2.1). Additionally, having each of the four aspects in the same paragraph suggest that they are invariably connected when this is not necessarily the case. The author is advised to rephrase this paragraph so as to remove any ambiguity/doubts surrounding the information being conveyed. Each point could be placed in its own paragraph.

In section 2.2, the author made a bold statement on "talented people governing the village..." This point important as it might be seemingly is void of any credible citation. Additionally, it begs the question as to where exactly this might be occurring. Would need revisiting by the author.

In the final paragraph of Section 2, the author in attempting to make a case in respect of "The nature of duties and powers... seemingly did not present a clear harmonization of thoughts. This made this point appear disjointed and out of place. This too warrants rephrasing in alignment with relevancy of the section it falls under.

Section 2.3, though a god showcase of the theoretical premise upon which the paper is written, again the author has been guilty of not citing correctly/not citing at all.

Response: We have made significant adjustments to this section, removing any inappropriate statements.

8.To this end the author is best advised to consider having a section in the manuscript devoted to the study population and giving us a clear background of what it is about the population that is special and warrants it being investigated by way of this study.

Response: We have added a subsection in the Methods section specifically dedicated to describing the study population (Study Subjects, lines 118-130).

9.Under section 3 (Research Design), it would augur well for good reading should the author include an area to address the study population specifically.

Response: The "Study Subjects" subsection in the second part of the Methods provides a detailed introduction to the research population.

10.Under section 3.2.2, it is felt that the first two sentences though a bold statement appear not to be cited. In the absence of citations here, the question asked is whether the author is presenting an opinion or a mere conjecture.

Response: We have removed the inappropriate statement.

11.Within the first paragraph of section 3.2.2, the author suddenly speaks of an econometric model and standardization, words of which connotes methods of analysis. Perhaps the author may opt to include a section that addresses specifically what these forms of analyses entail. Having a section specifically devoted to the methodology/methods employed could address this. This section could also include the kinds and types of analyses needed instead of morphing analysis with results.

Response: Thank you for your suggestion! This section includes variable definitions, and we acknowledge that we employed a straightforward approach. We have now consolidated this information in the Methods section.

12.Under section 3.2.3, again the author though referring to ''drawing on literature" fails to at the least cite one literature that that adopts the approach of controls for gender in the manner expressed in the manuscript. This is a n area for improvement by the author. the author is advised to at the least provide a basis/source for certain statements made or being alluded to.

Response: We have provided relevant literature to support the rationale for the selection of control variables.

13.Under section 4 the author appears to make reference to several kinds of analyses underpinning the results. However, what is not entirely clear is what are some of the key assumptions upon which these analyses are based. This need to be borne out in this section.

Response: In the "Baseline Estimation Model" subsection, we have explained the assumptions of the model. 

14.Additionally, re: paragraph that speaks of stepwise regression, the author in closing attempts to morph results with discussion/implications of work. The author is best advised to include this in the discussion/conclusion section of the manuscript.

Response: Additionally, we have moved the discussion of the stepwise regression results to the Discussion section (lines 238-246).

15.In section 4 (e.g section 4.4.1) of the manuscript, the author is often guilty of beginning to speak of a method of measurement/analysis but then in the same breath talks of a discussion. This disjointedness ought to be addressed. An explanation of a methodology cannot suddenly descend to a discussion of something else that appears unrelated.

Response: We have revised this subsection as requested. The methods-related content has been placed in Section 2, specifically in the "Data Analysis Methods" subsection, while the discussion-related content is now included in Section 4, the "Discussion."

16.The manuscript was void of page numbers and line numbering（）. This made the process of tagging areas of concern rather difficult for the reviewer.

Response: Thank you for your reminder. We have added page numbers and line numbers to the manuscript, and we apologize for any inconvenience this may have caused during your review.

17.The article title could be more specific/definitive and self-explanatory without having to refer to the body of the manuscript should the author choose to identify the study population therein.

Response: We have revised the paper’s title to make it more self-explanatory: "The Impact of Village Heads' Educational Levels on Adolescent Academic Performance: Evidence from Rural China."

18.The author is best advised to choose a single citation style instead of using two distinct citation styles at the same time. This flaw is very prominent in the introduction section of the manuscript. In other areas of the manuscript the author is guilty of alternating between citation styles. The author should seek to be consistent by using a single citation style.

Response: In accordance with the journal's requirements, we have standardized the reference style to "Vancouver."

*Reviewer 2

1.(e.g., abstract, introduction, methods, results, discussion, and conclusion). Adopting this format and streamlining the content could enhance overall clarity and coherence, reducing potential confusion for readers.

Response: Following your suggestion, we have restructured the manuscript to include the following framework: abstract, introduction, methods, results, discussion, and conclusion.

2.I recommend consistently using the term 'China Family Panel Studies' throughout the abstract and methods sections rather than using 'China Family Tracking Survey’ in the abstract.

Response: Thank you for pointing out the issue. We have made revisions to the abstract.

3.The literature review section is overly lengthy, and since this is not a systematic review, a concise summary would be more appropriate. I recommend merging the literature review with the introduction to create a focused and coherent introduction (specifically, 3 to 4 pages or less) that effectively summarizes the background, research context, and study hypothesis.

Response: We have combined the citations and literature review into a single section, and we hope the new version meets the standards of academic writing.

4.Table 1: Categorical variables, including gender, political identity, agricultural production, and internet use, should not be described using mean and standard deviation (SD), as these summary measures are appropriate only for continuous data with a normal distribution. Instead, categorical data should be presented using frequencies and percentages.

Response: We have corrected the previously erroneous expression (line 226).

5.Tables: I recommend naming the columns as Model 1, Model 2, and Model 3 etc., instead of (1), (2), and (3).

Response: We have standardized the terminology to use "Model 1" and "Model 2."

6.Tables: I understand that the significance levels of 1%, 5%, and 10% correspond to p-values of 0.01, 0.05, and 0.1, respectively. However, most research considers associations with p-values below 0.05 as statistically significant. Therefore, treating p-values of 0.1 as indicating a significant association may be unconventional and should be carefully justified.

Response: We have revised the expression of significance for all results. The manuscript consistently indicates that only results with p < 0.05 are considered significant. Additionally, we have provided explanations for the symbols * and ** in the footnotes of the tables.

7.To better understand the data, it is more appropriate to present the coefficients and their 95% confidence intervals.

Response: All results now include 95% confidence intervals, reported in brackets, for example, [0.0356, 0.226]. 

8.All tables should be self-explanatory. For example, in Table 5, Column 1: Edu_year 0.111***(0.023) – it's unclear what the numbers in brackets represent. Please clarify this in the table or the table notes.

Response: Explanations are provided at the bottom of all tables.

9.The conclusion is too lengthy. Including detailed discussions of mechanisms or extending the discussion detracts from the main points. This section should be concise, focusing exclusively on the study's major findings.

Response: We have revised the conclusion to ensure it is concise and focused on the main findings.

10.Policy recommendations: This section would be more effective if it informed the reader about the current policy and clearly explained how the study's findings could change, influence, or improve the existing policies. The current version lacks focus.

Response: We have emphasized the role of the "College Student Village Official" policy in China and how it can contribute to improvements in this area.

11.The manuscript contains a mixture of citation styles. I recommend using a single citation style consistently throughout the manuscript—either Vancouver or Harvard—according to the journal's guidelines.

Response: In accordance with the journal's requirements, we have standardized the reference style to "Vancouver."

12.The term 'highly educated' usually refers to individuals with advanced degrees, such as a PhD or master's, who have specialized knowledge and skills. Since village leaders in China are unlikely to hold such qualifications, I recommend using the term 'well-educated' throughout the manuscript instead. If the authors have a compelling justification for using 'highly educated,' they may retain it, but this should be clearly explained.

Response: Thank you for pointing out this issue; we have changed "highly educated" to "well-educated" throughout the manuscript.

13.I recommend restructuring the article to follow this sequence: abstract, introduction, 

---

## [Decision Letter · Decision Letter 1]

29 Oct 2024

PONE-D-24-18436R1The Impact of Village Heads' Educational Levels on Adolescent Academic Performance: Evidence from Rural ChinaPLOS ONE

Dear Dr. Li,

Thank you for submitting your manuscript to PLOS ONE. After careful consideration, we feel that it has merit but does not fully meet PLOS ONE’s publication criteria as it currently stands. Therefore, we invite you to submit a revised version of the manuscript that addresses the points raised during the review process.

Thank you for the revisions you have made. The reviewers still request some additional clarifications, and I would like the choice of instrument to be further justified by references to the literature. 

We look forward to receiving your revised manuscript.

Kind regards,

Leonard Moulin

Academic Editor

PLOS ONE

Journal Requirements:

Reviewers' comments:

Reviewer's Responses to Questions

**Comments to the Author**

1. If the authors have adequately addressed your comments raised in a previous round of review and you feel that this manuscript is now acceptable for publication, you may indicate that here to bypass the “Comments to the Author” section, enter your conflict of interest statement in the “Confidential to Editor” section, and submit your "Accept" recommendation.

Reviewer #1: All comments have been addressed

Reviewer #2: (No Response)

2. Is the manuscript technically sound, and do the data support the conclusions?

Reviewer #1: Yes

Reviewer #2: Partly

3. Has the statistical analysis been performed appropriately and rigorously? 

Reviewer #1: Yes

Reviewer #2: Yes

4. Have the authors made all data underlying the findings in their manuscript fully available?

Reviewer #1: Yes

Reviewer #2: Yes

5. Is the manuscript presented in an intelligible fashion and written in standard English?

Reviewer #1: No

Reviewer #2: Yes

6. Review Comments to the Author

Reviewer #1: The authors have done well in improving the structure and quality of the paper to include its coherence and layout among other things. Notwithstanding the improvements made, there yet remain a few more areas that should be tidied up or considered for improvement:

Line 26- author should relook at words 'well education levels. This does not read well. I take it that this might just be a typographical error.

Line 50- Authors may give consideration to linking rural areas of developing countries with the previous paragraph for the sake of coherence.

Line 58- Consider inserting a full stop after the word capabilities. Next sentence should start with 'Those"

Lines 111-115 -suggest that the study employed a quantitative research design, however from the last few sentences of the introduction one gets the impression that there were both qualitative and quantitative approaches involved. Authors need to clarify.

Line 76-106- Authors appear to be guilty of discussing implications of the study findings at this stage of the article. Authors may want to shift these elements to the discussion section of the paper and perhaps close the introduction section with just the rationale for the study.

Line 107-109 Not sure if this is warranted as the section head will bring this out.

Line 112- this paragraph could be restructured differently to eliminate the obvious repetitiveness in the presence of the second sentence. Not sure of the reason the author identifies quantitative assessment again when this was already stated in the first sentence.

Line 134 -authors suddenly speaks of 'survey questionnaire'. Are the authors saying that the official channels of CFPS uses the survey questionnaire as a tool. If this is correct then the authors should state this by using words such as " The survey questionnaire a tool used in the ....... includes....

This would make for better reading and lend to coherence.

Line 144- authors need to qualify the term basic characteristics. Are they referring to demographics, socio-demographics, clinical...what exactly?

Line 148- Authors may wish to reposition the ethical statement at the end of the Methods section instead of having t here.

Line 168- what was the approach used in standardizing test scores.

Line 197- Control variables. Are the authors saying that they controlled for confounding and that these were potential confounders. What exactly is the rationale for control variables.

Line 229- Can authors say what excatly is meant by conduct robustness checks. What was done exactly?

Lines 327-330: If this is relatable to what was stated in line 229, then authors would need to find a way to bring section together under methods. Would suggest that we not repeat methodology under results section. If we are reporting on results, then report on results and not repeat method.

Line 364-366: similar occurrence as in lines 327-330, where the authors in presenting the results of mechanism analysis appear to to be also describing the methods involve. Authors should review and ensure that only results are captured.

Line 394: why are the authors discussing or referencing discussions above when there is a section wholly devoted to discussions (line 415). Authors should review and make a decision on what should be in results and what should be under discussions.

Line 416, 437, 464: Authors may wish to refine these subheadings by taking out the word 'discussion' since the heading at line 415 already covers this. Further authors are asked to consider presenting the results in the format of

-findings (discuss all findings)

-discussions of findings in relation to existing literature

-strengths and limitations of the study

-implications of findings

This approach will significantly strengthen the paper and make for easy reading.

Line 494: Very good points, however, this section reads more like what the discussion section should read like or consist. The authors could tighten up the conclusion to produce at the most two significant paragraphs that speaks to key but impactful points emanating from the findings but are also overarching.

Reviewer #2: General Comment: I commend the authors for their efforts in revising the manuscript. The improvements made are commendable, and overall, the manuscript has been strengthened.

Comment #1:

In my previous review, I suggested adding 95% confidence intervals. In Model 2, the results now show: -0.106* [-0.230, 0.018]. The 95% confidence intervals indicate that the range includes zero, which means the result is not statistically significant at the 5% level. Please check if this is a typographical error and revise accordingly.

Comment #2:

In my previous feedback, I advised against extending the discussion into the conclusion section. It seems that this point was not fully addressed. For example, in line 511, there is a citation of other studies, which is not appropriate for this section. The Conclusion should not discuss findings or introduce new information.

7. PLOS authors have the option to publish the peer review history of their article (what does this mean?). If published, this will include your full peer review and any attached files.

Reviewer #1: No

Reviewer #2: **Yes: **Dr. Anderson Bendera

---

## [Author Response · Author response to Decision Letter 1]

8 Nov 2024

*Editor

1.Please review your reference list to ensure that it is complete and correct.. 

Response: We have conducted a thorough review of the references and have confirmed that none of the cited articles have been retracted.

*Reviewer 1

Thank you very much for your revision suggestions. These comments have significantly improved the clarity, accuracy, and precision of the manuscript. I will now address each of your suggestions in detail.

1.Line 26- Author should relook at words 'well education levels. This does not read well. I take it that this might just be a typographical error.

Response: In Line 26, we have revised the wording to "well-educated."

.

2.Line 50- Authors may give consideration to linking rural areas of developing countries with the previous paragraph for the sake of coherence. 

Response: Thank you for your suggestion. We have added a sentence at the beginning of the paragraph: "In light of these disparities, the role of village heads in rural areas of developing countries becomes increasingly significant" (Line 49). This addition helps to connect rural areas in developing countries to the previous paragraph.

3.Line 58- Consider inserting a full stop after the word capabilities. Next sentence should start with 'Those".

Response: In Line 57, we made modifications to improve the clarity of the language.

4.Lines 111-115 -suggest that the study employed a quantitative research design, however from the last few sentences of the introduction one gets the impression that there were both qualitative and quantitative approaches involved. Authors need to clarify.

Response: The previous version indeed had the potential to mislead readers. We have made the following modifications to the methodology section (Lines 85-87): "We used a cross-sectional study design to examine the association between village heads' educational levels and adolescents' academic performance, utilizing data from the CFPS to assess how these educational levels influence academic achievements." Additionally, we revised the introduction, specifically by removing the last few sentences.

5.Line 76-106- Authors appear to be guilty of discussing implications of the study findings at this stage of the article. Authors may want to shift these elements to the discussion section of the paper and perhaps close the introduction section with just the rationale for the study.

 Response: Thank you for your suggestion. We have revised the introduction for improved flow and clarity, and, following your advice, concluded it by emphasizing the fundamental principles of our study (Lines 68-82). The content that was previously more suited for discussion has now been relocated to the discussion section.

6.Line 107-109 Not sure if this is warranted as the section head will bring this out..

Response: We have removed the inappropriate statement.

7. Line 112- this paragraph could be restructured differently to eliminate the obvious repetitiveness in the presence of the second sentence. Not sure of the reason the author identifies quantitative assessment again when this was already stated in the first sentence.

Response: We have made significant adjustments to this section, removing any inappropriate statements (Lines 85-87). Specifically, we revised the text to: "We used a cross-sectional study design to examine the association between village heads' educational levels and adolescents' academic performance, utilizing data from the CFPS to assess how these educational levels influence academic achievements."

8.Line 134 -authors suddenly speaks of 'survey questionnaire'. Are the authors saying that the official channels of CFPS uses the survey questionnaire as a tool. If this is correct then the authors should state this by using words such as " The survey questionnaire a tool used in the ....... includes....This would make for better reading and lend to coherence..

Response: In Lines 105-108, we made the following modification: "The survey questionnaire, used as part of the CFPS, includes multiple-choice questions and scales that cover information on family background, educational environment, and characteristics of village heads."

9.Line 144- authors need to qualify the term basic characteristics. Are they referring to demographics, socio-demographics, clinical...what exactly?

Response: In Lines 113-115, we made the following modification: "The 2014 CFPS dataset includes basic characteristics, such as the economic and social conditions of 365 villages, as well as demographic information about the village heads, along with data on the academic performance of adolescents within these villages." This revision provides a clearer definition of the term "basic characteristics."

10.Line 148- Authors may wish to reposition the ethical statement at the end of the Methods section instead of having t here.

Response: We have repositioned the ethics statement to the end of the methodology section (Lines 270-276).

11.Line 168- what was the approach used in standardizing test scores.

Response:My apologies for the oversight. We have standardized the test scores using z-scores based on the children's current educational level. In Lines 130-132, we provided the following explanation: "However, more educated children might achieve higher scores, so this paper standardizes the test scores based on the children's current educational level using z-scores to reduce intergroup differences."

12.Line 197- Control variables. Are the authors saying that they controlled for confounding and that these were potential confounders. What exactly is the rationale for control variables.

Response: We apologize for any confusion caused. The term "control variable" is used in economics and corresponds to "covariate." We have revised it to "covariates." The selection of covariates was based on a priori theoretical considerations. The related modifications can be found in Lines 160-185.

13.Line 229- Can authors say what excatly is meant by conduct robustness checks. What was done exactly? Lines 327-330: If this is relatable to what was stated in line 229, then authors would need to find a way to bring section together under methods. Would suggest that we not repeat methodology under results section. If we are reporting on results, then report on results and not repeat method.

Response: In our robustness checks, we aimed to verify the stability and reliability of our results by using alternative model specifications and different sample subsets. We have moved the content originally in Lines 327-330 to the methodology section. This part explains why we conducted the robustness checks and specifically what methods were used (Lines 229-244).

14.Line 364-366: similar occurrence as in lines 327-330, where the authors in presenting the results of mechanism analysis appear to to be also describing the methods involve. Authors should review and ensure that only results are captured.

Response: We have removed this inappropriate statement.

15.Line 394: why are the authors discussing or referencing discussions above when there is a section wholly devoted to discussions (line 415). Authors should review and make a decision on what should be in results and what should be under discussions..

Response: We apologize for the confusion. Our initial intention was to inform readers about the purpose of this section, but the wording was problematic. We have now removed this part in the current version.

16.Line 416, 437, 464: Authors may wish to refine these subheadings by taking out the word 'discussion' since the heading at line 415 already covers this. Further authors are asked to consider presenting the results in the format of

-findings (discuss all findings)

-discussions of findings in relation to existing literature

-strengths and limitations of the study

-implications of findings

This approach will significantly strengthen the paper and make for easy reading..

Response: Thank you for your suggestion. We have now streamlined the discussion section, ensuring that each research conclusion is discussed in a separate paragraph. Additionally, we structured our discussion according to the following components: findings, discussion of findings in relation to existing literature, strengths and limitations of the study, and implications of the findings. The related modifications can be found in Lines 380-450.

17.Line 494: Very good points, however, this section reads more like what the discussion section should read like or consist. The authors could tighten up the conclusion to produce at the most two significant paragraphs that speaks to key but impactful points emanating from the findings but are also overarching.

Response: In the current version, we have further condensed the conclusion section. We summarized the key findings and their significance while emphasizing directions for future research (Lines 452-464).

*Reviewer 2

Thank you for pointing out this issue. We have carefully reviewed all statistical analyses to ensure that similar issues do not occur, and we have also made revisions to the conclusion section. I will now address each of your suggestions in detail.

1.In my previous review, I suggested adding 95% confidence intervals. In Model 2, the results now show: -0.106* [-0.230, 0.018]. The 95% confidence intervals indicate that the range includes zero, which means the result is not statistically significant at the 5% level. Please check if this is a typographical error and revise accordingly.

Response: We sincerely apologize for the formatting oversight. This was an error, and it should be consistent with the significance level reported in Model 3. We have corrected this issue (Line 281) based on your comment regarding this problem.

2.In my previous feedback, I advised against extending the discussion into the conclusion section. It seems that this point was not fully addressed. For example, in line 511, there is a citation of other studies, which is not appropriate for this section. The Conclusion should not discuss findings or introduce new informationt.

Response: We have rewritten the conclusion, focusing only on the main findings of the study and directions for future research (Lines 452-464).

---

## [Decision Letter · Decision Letter 2]

27 Nov 2024

PONE-D-24-18436R2The Impact of Village Heads' Educational Levels on Adolescent Academic Performance: Evidence from Rural ChinaPLOS ONE

Dear Dr. Li,

Thank you for submitting your manuscript to PLOS ONE. After careful consideration, we feel that it has merit but does not fully meet PLOS ONE’s publication criteria as it currently stands. Therefore, we invite you to submit a revised version of the manuscript that addresses the points raised during the review process.

We look forward to receiving your revised manuscript.

Kind regards,

Leonard Moulin

Academic Editor

PLOS ONE

Journal Requirements:

Additional Editor Comments:

I sent your paper back out to the original referees. They are pleased with the revisions and suggest that I now accept the paper for publication. I agree that the paper has moved in the right direction. I do have one additional comment that I would like to see addressed in a further revision: I would like the choice of instrument to be further justified by references to the literature. Once you provide this I expect to make a final decision.

Reviewers' comments:

Reviewer's Responses to Questions

**Comments to the Author**

1. If the authors have adequately addressed your comments raised in a previous round of review and you feel that this manuscript is now acceptable for publication, you may indicate that here to bypass the “Comments to the Author” section, enter your conflict of interest statement in the “Confidential to Editor” section, and submit your "Accept" recommendation.

Reviewer #1: All comments have been addressed

Reviewer #2: All comments have been addressed

2. Is the manuscript technically sound, and do the data support the conclusions?

Reviewer #1: Yes

Reviewer #2: (No Response)

3. Has the statistical analysis been performed appropriately and rigorously? 

Reviewer #1: Yes

Reviewer #2: (No Response)

4. Have the authors made all data underlying the findings in their manuscript fully available?

Reviewer #1: Yes

Reviewer #2: (No Response)

5. Is the manuscript presented in an intelligible fashion and written in standard English?

Reviewer #1: Yes

Reviewer #2: (No Response)

6. Review Comments to the Author

Reviewer #1: The authors have since addressed my comments. To this end I am comfortable that the manuscript has been further strengthened.

Commendations to the authors for their perseverance and commitment to the process of peer review.

Except for two typographical errors that could be easily addressed at editing for publication re: lines 85 -word separation for across-sectional to " a cross-sectional" and lines 386-387 where the author need to insert their citation at the end of the sentence on "Karadag's..." I find no other issue with manuscript.

Reviewer #2: (No Response)

7. PLOS authors have the option to publish the peer review history of their article (what does this mean?). If published, this will include your full peer review and any attached files.

Reviewer #1: No

Reviewer #2: **Yes: **Anderson Bendera

---

## [Author Response · Author response to Decision Letter 2]

29 Nov 2024

Thank you for your positive feedback regarding our revisions, and for the opportunity to further improve the manuscript. We are grateful for the valuable comments provided by both you and the referees, and we appreciate the detailed consideration given to our work.

Regarding your additional comment about the justification for the choice of instruments, we have made the following clarifications and revisions:

First, concerning the choice of research instruments, we have added references to substantiate our use of the China Family Panel Studies (CFPS) survey data. Specifically, we have cited relevant literature demonstrating that the CFPS is widely used and accepted in academic research, particularly in economic studies. For instance, research utilizing CFPS data has been published in leading economics journals such as the American Economic Review, and articles in PLOS ONE have also utilized CFPS data to support their findings. These references are now included in lines 109 and 110 to emphasize the robustness and validity of using CFPS data in our study.

Second, we understand that there may have been concerns regarding the instrumental variables employed in our analysis. To address this, we have expanded on the rationale for the selection of our instrumental variables. In econometric and social science research, instrumental variable (IV) methods are commonly used to mitigate endogeneity issues, thereby enhancing the accuracy of the study's conclusions. Specifically, we used community-level averages as instruments for individual-level variables, which is a standard approach in the field. We have supported this choice by including citations from authoritative sources, including the work of the renowned econometrician Jeffrey Wooldridge, to further substantiate our methodological choices. These additions can be found in lines 218-230 of the revised manuscript.

In addition, we have addressed the two errors that reviewer pointed out. Specifically, we have corrected the word separation in line 85 from "across-sectional" to "a cross-sectional". We have also inserted the appropriate citation at the end of the sentence on "Karadag's..." in lines 396.

We hope that these additions and clarifications address your concerns satisfactorily. Additionally, we acknowledge that we may have misunderstood your reference to the 'choice of instrument', but we are confident that all the research instruments used in the manuscript are reliable. We are happy to provide any further revisions or additional details if needed. Once again, we greatly appreciate your constructive feedback and the opportunity to finalize this manuscript for publication.

Best regards,

Jing Li

Huan Deng

Jun Li

---

## [Editor Report · Decision Letter 3]

2 Dec 2024

The Impact of Village Heads' Educational Levels on Adolescent Academic Performance: Evidence from Rural China

PONE-D-24-18436R3

Dear Dr. Li,

We’re pleased to inform you that your manuscript has been judged scientifically suitable for publication and will be formally accepted for publication once it meets all outstanding technical requirements.

Kind regards,

Leonard Moulin

Academic Editor

PLOS ONE
---

## [Editor Report · Acceptance letter]

18 Dec 2024

PONE-D-24-18436R3 

PLOS ONE

Dear Dr. Li, 

I'm pleased to inform you that your manuscript has been deemed suitable for publication in PLOS ONE. Congratulations! Your manuscript is now being handed over to our production team.

Kind regards, 

on behalf of

Dr. Leonard Moulin 

Academic Editor

PLOS ONE